# Silver Diamine Fluoride (SDF) Efficacy in Arresting Cavitated Caries Lesions in Primary Molars: A Systematic Review and Metanalysis

**DOI:** 10.3390/ijerph191912917

**Published:** 2022-10-09

**Authors:** Luciano Zaffarano, Claudia Salerno, Guglielmo Campus, Silvia Cirio, Araxi Balian, Lorena Karanxha, Maria Grazia Cagetti

**Affiliations:** 1Department of Biomedical, Surgical and Dental Sciences, University of Milan, Via Beldiletto 1, 20142 Milan, Italy; 2Department of Restorative, Preventive and Pediatric Dentistry, University of Bern, Freiburgstrasse 7, 3012 Bern, Switzerland; 3Department of Pediatric, Preventive Dentistry and Orthodontics, School of Dentistry, Sechenov University, 119991 Moscow, Russia; 4Department of Surgery, Microsurgery and Medicine Sciences, School of Dentistry, University of Sassari, Viale San Pietro 3/c, 07100 Sassari, Italy

**Keywords:** silver diamine fluoride, SDF, dental caries, primary teeth, non-invasive treatment, minimally invasive treatment

## Abstract

A systematic review and meta-analysis were carried out to evaluate the efficacy of silver diamine fluoride (SDF) in controlling caries progression in cavitated primary molars. A search for randomized and non-randomized trials with follow-up > 6 months was performed using PubMed, Scopus and Embase. The Cochrane risk of bias tools were used for the quality assessment. The success rate and odds ratios were chosen to calculate the effect size for the meta-analysis. A total of 792 papers were identified and 9 were selected. A high variability regarding SDF application protocol was found; otherwise, caries arrest was always recorded using visual/tactile methods. Two studies were judged at low risk of bias, six at moderate risk and one at high risk. Data from five studies were aggregated for meta-analysis. Heterogeneity was found moderate (I^2^ = 35.69%, *p* = 0.18). SDF application was found to be overall effective (fixed effect model) in arresting caries progression (ES = 0.35, *p* < 0.01). In a total of 622 arrested lesions, out of 1205 considered, the caries arrest rate was 51.62% ± 27.40% (Confidence = 1.55) using SDF ≥ 38% applied annually or biannually. In conclusion, when applied to active cavitated caries lesions in primary molars, SDF appears to be effective in arresting dental caries progression, especially if applied biannually.

## 1. Introduction

More than 570 million children around the world present with untreated caries lesions, and these data have unfortunately remained almost constant over the last decades [1,2]. Caries of primary teeth ranks second among non-communicable diseases in children aged 0–14 years, according to the 2019 Global Burden of Disease (prevalence data from the Global Health Data Exchange tool) [3].

The caries progression rate in primary dentition is higher than in permanent dentition, due to their anatomical and histological differences [4]. As primary molars should last longer than anterior teeth, their treatment is one of the most frequent challenges in the daily routine of pediatric dentists. Toddlers and pre-school children are not always sufficiently cooperative in conventional “surgical” procedures. For this reason, simple and painless caries management, as well as personalized behavioral approaches, are preferable [5,6]. As a consequence, non-invasive or minimally invasive therapeutic approaches are strongly recommended in children [7,8]. The non-invasive treatment consists of the promotion of remineralization and the modification of the biofilm at lesion level, arresting the progressive loss of residual dental tissue [9,10]. Fluoride is the essential component of the non-invasive approach [11], with regards not only to caries lesions prevention, but also to the control and arrest of the progression rate both at enamel and dentin stages [12]. Highly concentrated fluoride-based compounds, such as silver diamine fluoride (SDF) solutions or fluoride varnish, showed a good efficacy for this strategy [13,14].

Silver diamine fluoride is a solution based on silver nitrate and fluoride, acting both as an anti-bacterial and a remineralizing agent. Reducing biofilm counts and hindering the degradation of collagen, the product favors the remineralization of enamel and dentin, thus arresting caries progression [15]. SDF has been used for decades since the late 1960s in Japan, China, Brazil and Argentina, at concentrations ranging from 10% to 38% [16,17,18,19,20]. In 2014, after its approval by the Food and Drug Administration as a desensitizing agent, its use and scientific interest exploded in the United States and recently in Europe. [21]. SDF is an alternative treatment for controlling dental caries when other approaches are not available. It is a minimally invasive, low-cost and simple method that can reduce fear and anxiety in young children. In addition, it could be applied in community settings [17].

Systematic reviews and meta-analyses are available in literature describing the effectiveness of SDF in preventing and arresting active caries lesions in primary teeth [22,23,24,25,26]. When applied to caries lesions in primary teeth, SDF compared to no treatment, placebo or fluoride varnish appears to effectively prevent dental caries in the entire dentition [22]. No differences in caries arrest were found between SDF and restorative procedures with glass ionomer cements (GIC) [22,23]. However, no previous review has investigated its effectiveness limited to cavitated posterior primary teeth and no appropriate application frequency has been defined; consequently, the outcomes from the systematic reviews are significantly biased as molar caries treatment is more difficult than that for anterior caries. This may have a negative prognostic effect on any type of intervention, even non-invasive approaches [27].

Starting from this premise, a systematic review was planned and carried out to verify the effectiveness of SDF in arresting active cavitated caries lesions in primary molars compared to no treatment or any other type of non-invasive or minimally-invasive treatment.

## 2. Materials and Methods

This systematic review follows the Preferred Reporting Items for Systematic Reviews and Meta-Analyses (PRISMA) guidelines [28], and it was registered on PROSPERO (registration number CRD42021259150).

The PICO model was used to structure the clinical research question by defining the inclusion criteria [29]. Thus, the present review aimed to systematically retrieve and analyze clinical studies investigating the efficacy of SDF in arresting active cavitated caries lesions in primary molars compared to no treatment or any other type of non-invasive or minimally-invasive treatment.

Population: children with active dentin cavitated lesions in primary molars;Intervention: silver diamine fluoride;Comparison: no treatment or any other type of non-invasive or minimally-invasive treatment;Outcome: caries arrest rate in different timeframes (primary outcome); patient’s discomfort during the treatment procedure (secondary outcome).

### 2.1. Eligibility Criteria

The inclusion criteria were:Type of study: randomized (RCT) and non-randomized clinical studies (NRSI);Publication languages: papers published in English, Italian and French;Time of publication: no time restriction applied, last accessed on 10 January 2022;Type of intervention applied: SDF applied in active dentin cavitated lesions in primary molars (first and second molars);Follow up: longer than 6 months;Primary outcome: caries arrest rate, mean number of inactivated lesions, odds ratios;Secondary outcome: patient’s discomfort during the treatment procedure.

### 2.2. Information Sources and Search Strategy

Three electronic databases, PubMed, Embase and Scopus were searched from the inception until 10 January 2022 by four authors (C.S., G.C., L.Z. and A.B.). The search strategy included a search string for each database.

For PubMed, the string used was: (“dental caries” [MeSH Terms] OR “dentin caries” [Title/Abstract] OR “dental cavity” [Title/Abstract] OR “caries arrest rate” [Title/Abstract] OR “caries activity” [Title/Abstract] OR “caries progression” [Title/Abstract] OR “tooth, deciduous” [MeSH Terms] OR “carious lesion*” [Title/Abstract] OR “caries lesion*” [Title/Abstract] OR “deciduous dentition” [Title/Abstract] OR “primary dentition” [Title/Abstract] OR “primary teeth” [Title/Abstract] OR “primary tooth” [Title/Abstract]) AND (“silver fluoride”[Supplementary concept] OR “silver diamine fluoride”[Supplementary concept] OR SDF[Title/Abstract] OR “silver fluoride”[Title/Abstract] OR “diamine fluoride*”[Title/Abstract] OR “diammine”[Title/Abstract] OR “silver nitrate solutions”[Title/Abstract]).For Embase: (‘dental caries’/exp OR ‘dental caries’ OR ‘dentin’/exp OR ‘dentin’ OR ‘dental cavities’/exp OR ‘dental cavities’ OR ‘caries progression’ OR ‘caries arrest’ OR ‘deciduous tooth’/exp OR ‘deciduous tooth’ OR ‘primary dentition’/exp OR ‘primary dentition’) AND (‘silver fluoride’ OR ‘silver diamine fluoride’ OR ‘silver diammine’ OR ‘silver nitrate’) AND ‘article’/it AND ‘human’/de NOT ‘in vitro study’/de AND [child]/lim.For Scopus: (TITLE-ABS-KEY (dental AND caries) OR TITLE-ABS (dentin AND caries) OR TITLE-ABS (dental AND cavity) OR TITLE-ABS (caries AND arrest) OR TITLE-ABS (caries AND progression) OR TITLE-ABS (caries AND activity) OR TITLE-ABS-KEY (deciduous) OR TITLE-ABS-KEY (carious AND lesion) OR TITLE-ABS-KEY (caries) OR TITLE-ABS-KEY (primary AND dentition) OR TITLE-ABS-KEY (primary AND teeth) OR TITLE-ABS-KEY (primary AND tooth)) AND (TITLE-ABS-KEY (silver AND fluoride) OR TITLE-ABS-KEY (sdf) OR TITLE-ABS-KEY (silver AND diamine AND fluoride) OR TITLE-ABS-KEY (silver AND fluoride) OR TITLE-ABS-KEY (diamine AND fluoride) OR TITLE-ABS-KEY (diammine) OR TITLE-ABS (silver AND nitrate AND solution)) AND (LIMIT-TO (DOCTYPE, “ar”)) AND (LIMIT-TO (SUBJAREA, “DENT”)).

Cross-referencing was also performed using the references lists of full-text articles. Gray literature was also retrieved via opengrey.eu (http://www.opengrey.eu).

### 2.3. Study Selection

The outputs of the reference searches were uploaded into a spreadsheet (Microsoft Excel^®^), and duplicates were removed. Two authors (S.C. and C.S.) examined all papers independently by title and abstract, and papers meeting the inclusion criteria were obtained in the full-text format. The same authors assessed the papers to establish whether each paper should or should not be included in the systematic review. Disagreements were resolved through discussion and/or by full-text analysis in doubtful cases. Where resolution was not possible, another author was consulted (M.G.C.).

### 2.4. Data Collection

Data collection and synthesis were independently performed by two authors (C.S. and S.C.) using an ad-hoc-designed data extraction form (Appendix A), without masking the name of the journal, title or authors. Numerical data were extracted and rounded up to two decimals; if this was not possible, data were extracted as they were reported by included papers.

### 2.5. Risk of Bias

No NRSI studies met the inclusion criteria, so only the Cochrane collaboration’s RoB 2 tool was used [30]. The risk of bias assessment was evaluated independently by three reviewers (C.S., G.C. and S.C.) and then discussed together with a fourth reviewer (M.G.C.) in order to resolve disagreements and provide the overall final judgment for each study.

A list of criteria to be followed to assess bias in each domain was agreed among the four authors. The SDF application protocol is not standardized, and each brand recommends different application modalities. This was therefore not considered for the bias assessment. A list of confounding domains and co-interventions were established and identified as: type and severity of caries lesions; age and outcome assessment for different types of treatment; presence of a control group. Bias related to deviation from the treatment protocol were rated as low, if SDF application was administered by dental personnel; as moderate, if it was administered by health personnel; as serious/critical, if the treatment was provided by non-medical or untrained personnel. Drop-outs were judged as follows: drop-outs lower than 10%, low risk; drop-outs of 10–20%, moderate risk; drop-outs of 20–30%, serious risk; drop-outs more than 30%, critical risk. Since SDF involves tooth discoloration, while other treatments do not or do sometimes involve cavity closure, it is impossible to maintain blinding at the follow-up evaluations. For this reason, the authors have decided to modify the final decision regarding blinding, reducing the impact that this domain would have had on the overall final judgment.

Microsoft Excel^®^ tool for RoB 2 was used to input answers given to signaling questions, and an algorithm estimated the overall risk of the bias according to the results for each domain as: low risk, some concerns, or high risk; a plot was then drawn using the Cochrane RoBvis web app [31].

### 2.6. Statistical Analysis

Inter-authors reliability was assessed as percentage of agreement using Cohen’s Kappa statistics.

Prometa3 Software (Internovi, Borne, The Netherlands, 2015) was used for the meta-analysis. The success rate (SR) and odds ratios (OR) were chosen to calculate the efficacy of the treatment. The analysis was computed on the different types of treatment used. A meta-analysis model was run if two or more studies compared the effect of SDF in arresting active dentin-cavitated lesions in primary molars versus no treatment or any other type of non-invasive or minimally invasive treatment, using comparable outcomes (G.C. and S.C.).

The primary outcome measured was the caries arrest rate. When there were more than one relevant intervention and/or comparison groups, they were combined into different subgroups, as recommended in the Cochrane Handbook for Systematic Reviews of Interventions [32].

Confidence was calculated through Filler’s method using Microsoft Excel^®^. The estimate of the between-studies variance under the random-effects model has poor precision when the number of studies is very small. For this reason, the fixed-effect model and the inverse variance method to obtain pooled estimates of caries arrest rates were used [33]. The I^2^ statistics were calculated to describe the percentage of variation across studies due to heterogeneity rather than chance [34]. The heterogeneity was categorized as follows: <30%, not significant; 30–50%, moderate; 51–75%, substantial, and 76–100%, considerable.

## 3. Results

### 3.1. Study Selection

The search strategy detected 796 papers and, after removing any duplicates, 514 were selected. Of these, after evaluating titles and abstracts, 485 papers were excluded with 84.00% agreement between raters (Cohen’s k = 0.29; Appendix A). The remaining 29 papers were obtained in the full-text format (Figure 1) and were assessed; 18 were discarded with a percentage of agreement of 97.53% between the raters (Cohen’s k= 0.93; Appendix A).

Nine studies were finally included in this systematic review [35,36,37,38,39,40,41,42,43,44]. From the ten studies initially considered eligible, one was excluded due to bias in data reporting (Table 1) [44]. All the papers, except one [35], were published between 2018 and 2021.

### 3.2. Subjects Involved

Overall, 3168 children were included with an age range ranging from 1 to 10 years. Regarding caries classification, eight papers [35,37,38,39,40,41,42,43] used the dmft index, one the dmfs [36], while the other two papers used both the dmft and the ICDAS [35,41]. The caries indices showed dmft ranging from 3.55 to 6.72 and ICDAS higher than 4. Only one study did not use a standardized index of caries lesions [36]. Three studies also considered the visible plaque index at baseline [37,39,42], and this was always found to be greater than 60%.

### 3.3. Study Characteristics

The summary of the nine included studies is displayed in Table 2 (studies included in meta-analysis, n = 5) and Table 3 (studies not included in meta-analysis, n = 4).

All the studies included in the review were RCTs. Six were double-arm trials [35,38,39,40,41,42] and three were more than two-arm trials [36,37,43]. Four papers had a sample size greater than 300 participants, [37,38,39,42] four studies lasted more than 12 months, [37,38,42,43] with a follow-up ranged from 6 to 36 months. One study compared 38% SDF to 5% sodium fluoride varnish [39]; three compared SDF to ART (atraumatic restorative treatment) [35,41,43]; one study compared 38% SDF to 12% SDF [37]; two studies compared 38% SDF to different silver-based varnishes [40,42]; one compared 38% SDF to no treatment [38]; one study compared 38% SDF different application protocols [36]. The interval time of SDF application ranged in the different trials, as described in Table 2 and Table 3. Only one study assessed patient discomfort during the treatment procedure (SDF versus ART) through the patient’s anxiety scale, obtaining almost overlapping results (*p* = 0.15) [41]. Eight studies used 38% SDF; one study used 30% SDF [41].

### 3.4. SDF Application Protocol

A high variability regarding SDF application protocol was observed. One study did not describe the application protocol at all [37]. As for the pre-operative procedures, in seven studies [35,36,38,39,40,41,42] no attempt was made to remove caries tissue from the affected primary teeth, while only in one study the superficial soft caries tissue was removed using hand instruments [43]. Six studies reported isolation procedures [35,36,38,39,40,41].

In six studies, SDF was applied directly on the affected teeth with a micro-sponge, allowing it to soak the solution for 5 s to 3 min with or without air-drying [35,36,38,39,41,42]. In one study, the solution was applied for 10 s followed by cavity closure with a cotton pellet for 10 min [40]. In the last two studies, no information was given regarding the application procedure [37,43].

In five studies, children were instructed to avoid eating or drinking for at least 30 min after the application [36,39,40,42,43].

In five studies, a second application was administrated after 6 months [35,38,39,42,43]; in two studies, additional applications varied according to different group protocol, from monthly to annually [36,37]. In two studies, no second application was provided [40,41].

In addition, the protocol setting was mixed; SDF was applied by dentists in seven papers [35,37,38,40,41,42,43], three of which in a hospital environment [35,41,43], four in a non-hospital environment [37,38,40,42], one by nurses in a non-hospital environment [39] and in a kindergarten, but it was not specified by whom [36].

The caries arrest was recorded using a visual and tactile evaluation. If a carious lesion resulted soft when probing with a light force, this was classified as active; if it resulted hard, it was considered arrested.

No studies used potassium iodide after SDF application.

### 3.5. Risk of Bias Assessment

Some papers reported that an intention-to-treat analysis was performed, but data were not available [35,39,41,42]. A per-protocol analysis was therefore conducted for all studies with the aim of assessing the effect of starting and adhering to the intervention. Two studies were judged at a low risk of bias [37,42]; six were judged at moderate risk [35,38,39,40,41,43], and one at high risk [36]. Bias arising from the measurements of the outcomes and the selection of the reported results highly affected the quality rating of the studies (Figure 2) [35,38,41].

### 3.6. Meta-Analysis

Data from five studies (Table 2) were aggregated for the meta-analysis [35,37,39,40,41]. Heterogeneity was found to be moderate (I^2^ = 35.99% *p* = 0.18). SDF application was found to be overall effective in preventing caries progression (effect size= 0.33 _95%_CI = 0.26–0.40, *p* < 0.01) (Figure 3a and Appendix A). In total, 622 arrested lesions out of the 1205 considered, were recorded. The total caries arrest rate was 51.62% ± 27.40% with a confidence of 1.55 using SDF ≥ 30% applied annually or biannually evaluated with a follow up major or equal to 12 months. The effect size using SDF 38% biannually ranged from 0.38 to 0.85 [35,39], while with an annual application, effect size ranged from −0.13 to 0.28 (Figure 3b) [37,40]. The moderator analysis for the type of treatment (SDF vs ART) was not statistically significant (Figure 3c), showing a higher effect size for SDF compared to ART.

## 4. Discussion

The clinical effectiveness of the SDF to halt caries progression and to increase remineralization has been extensively studied in the last decade [14,15].

Although good efficacy has been reported from clinical studies and confirmed by reviews, a different efficacy between anterior and posterior teeth has been shown in some clinical studies [42,43]. Furthermore, as posterior teeth remain in the mouth for a longer time and are at greater risk of caries as they are more difficult to clean and more susceptible to plaque, it was necessary to investigate efficacy in these teeth and not overall efficacy [45].

However, no systematic review has been conducted to analyze SDF efficacy in arresting caries process in cavitated primary molars nowadays [22,24]. The results of the present meta-analysis demonstrate that SDF is effective in arresting cavitated lesions in primary molars, especially when applications are repeated biannually, even though the overall scientific evidence is still poor. The total caries arrest rate found in this meta-analysis is lower than those reported by previous meta-analyses, in which both posterior and anterior primary teeth were considered [22,24]. Bearing in mind that SDF has greater efficacy on anterior teeth, the results reported in the present paper are encouraging. Several papers were excluded from the review as they did not provide results differentiated between anterior and posterior teeth. In addition, two studies were excluded from the meta-analysis, as the data available for the posterior teeth were only partially reported [42,43], demonstrating how good quality studies on SDF efficacy on posterior teeth are still lacking.

Among the critical aspects in the study design of the selected papers, there is the use of the dmft to assess caries figure. While the index is still a valid resource for epidemiological purposes, it doesn’t provide caries severity information as it is only dichotomous (presence/absence of caries). ICDAS II seems to be a more adequate index [46] to analyze caries increment in severity; however, it was used only in two studies [35,41].

The lack of standardization of the SDF protocol application emerged from all studies, potentially affecting their outcomes. In addition, the setting procedures differ among the studies; i.e., if an effective isolation is not obtained, the effectiveness of the therapy can be impaired or limited [41].

Despite the strong variability observed in the application protocols adopted, the meta-analysis shows that SDF is more effective compared to fluoride varnish or ART technique, although less effective than other silver-based compounds [40]. Repeated applications (at least every six months) are more effective in caries arrest, calling for the need of an evidence-based application protocol [39,41].

Another interesting result is that, when SDF and ART are compared [41], the SDF might be more effective in the short-term, however, if other applications are not carried out, its effectiveness decreases. It is therefore possible to speculate that treatment with SDF is effective if it is not used as a single-shot treatment [36]. In fact, SDF applied biannually was more effective than ART, but the overall effect size found was not statistically significant.

The included papers adopted variable protocols, both in terms of application modalities and in terms of number of applications over time, consequently influencing the effectiveness of the solution. Therefore, more thorough studies analyzing the efficacy of SDF in cavitated lesions of the posterior primary teeth with uniform application modalities and timing are needed to further increase the strength of the recommendation. This paper referred exclusively to SDF use in posterior teeth of primary dentition. The posterior teeth enjoy less benefit by the use of fluoride-based treatments due to their anatomical position, greater difficulties in obtaining adequate isolation and more uncomfortable toothbrushing [41].

Obtaining reliable data to support the use of SDF treatment in primary molars could provide crucial help for clinicians, as toddlers and preschool children have a low level of collaboration to traditional surgical procedures.

## 5. Conclusions

SDF 38% was found to be effective in arresting active cavitated lesions in primary molars when applied annually or biannually, evaluated at follow up ≥12 months. SDF tends to be more effective than ART technique or NaF varnish in arresting caries progression. The effectiveness of SDF greatly increases if applications are repeated over time. A lack of standardization in the application protocol emerged from the literature.

## Figures and Tables

**Figure 1 ijerph-19-12917-f001:**
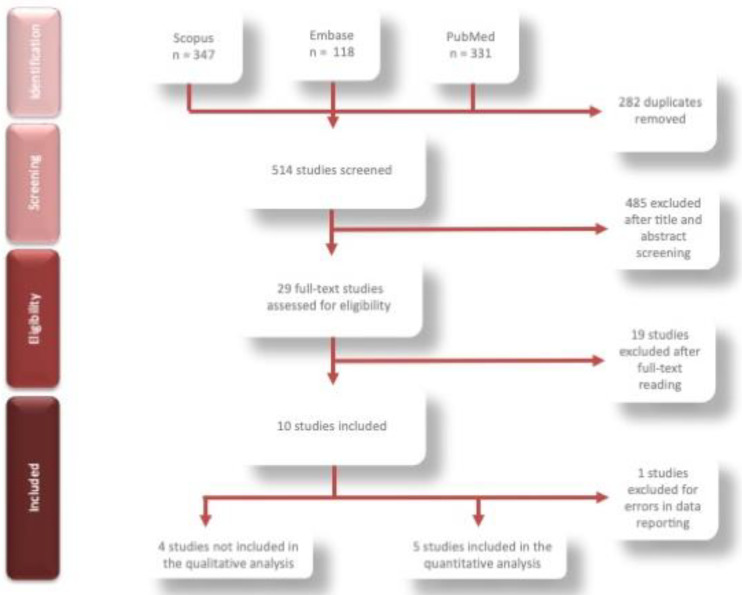
Flowchart of the search strategy and process of the identification of the papers included.

**Figure 2 ijerph-19-12917-f002:**
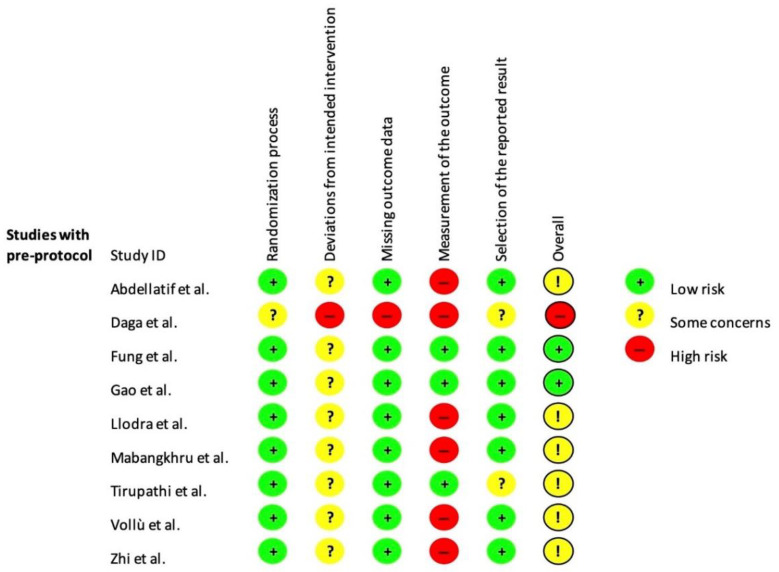
Risk of bias assessment of included studies using the RoB 2 [35,36,37,38,39,40,41,42,43].

**Figure 3 ijerph-19-12917-f003:**
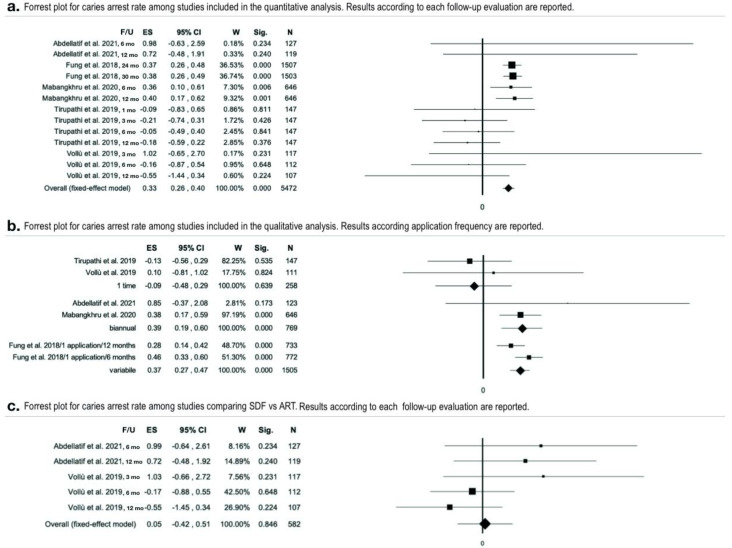
Metanalyses [35,37,39,40,41]. (**a**) Forest plot for caries arrest rate. Results for each study (fixed-effect model) are combined for subgroups but not for follow-ups; (**b**) forest plot for caries arrest rate comparing different application frequency. Moderator analysis (application as categorical moderator), overall forest plot (fixed-effect model) combined for follow-up; (**c**) forest plot for caries arrest rate among studies comparing SDF and ART (fixed-effect model) combined for subgroups but not for follow-ups.

**Table 1 ijerph-19-12917-t001:** General characteristics of the included studies regarding the efficacy of SDF in arresting caries lesions in primary molars.

Authors (Year)	Sources	Location	Database	Type of Study
Abdellatif et al., 2021 [35]	*Eur Arch Paed Dent*	Saudi Arabia	Pubmed, Embase, Scopus	RCT
Zhi et al., 2021 [43]	*J Dent*	China	Pubmed, Embase, Scopus	RCT
Gao et al., 2020 [42]	*J Dent*	China	Pubmed, Embase, Scopus	RCT
Mabangkhru et al., 2020 [39]	*J Dent*	China	Pubmed, Embase, Scopus	RCT
Daga et al., 2020 [36]	*Indian J Public Health Res Dev*	India	Embase	RCT
Tirupathi et al., 2019 [40]	*J Clin Exp Dent*	India	Scopus	RCT
Vollú et al., 2019 [41]	*J Dent*	Brazil	Pubmed, Embase, Scopus	RCT
Fung et al., 2018 [37]	*J Dent Res*	China	Pubmed, Scopus	RCT
Llodra et al., 2005 [38]	*J Dent Res*	Spain	Pubmed, Embase, Scopus	RCT
RCT = Randomized Controlled Trial				

**Table 2 ijerph-19-12917-t002:** Main characteristics of the studies included in meta-analysis (results rounded up to 1 decimal when 2 decimals were not available).

Authors (Year)	N-Participants (% Dropout)		Age-Range	M/F	Caries Prevalence at Baseline	Caries Lesions in Primary Molars at Baseline (n)		Intervention		Blinding	Findings (%)		
	Test	Control				Test	Control	Test	Control			Test	Control
Abdellatif et al. (2021) [35]	40 † (32.50%)	39 † (33.33%)	3–8 yy †	32/47 †	dmft = 4.13 †	59	98	38% SDF at baseline and every 6 mo.	ART	No	6 mo.	100.00% (49/49)	94.87% (74/78)
											12 mo.	97.95% (48/49)	92.86% (65/70)
					ICDAS †								
					4 = 13.9%								
					5 = 62.4%								
					6 = 23.6%								
Mabangkhru et al. (2020) [39]	153 † (15.03%)	149 † (10.73%)	1–5 yy †	172/130 †	dmft = 5.27 †	279	367	38% SDF at baseline repeated every 6 mo.	5% NaF varnish at baseline repeated every 6 mo.	Participants and examiner	6 mo.	17.20% (48/279)	9.81% (36/367)
											12 mo.	23.65% (66/279)	13.08% (48/367)

Tirupathi et al. (2019) [40]	26 (7.69%)	24 (4.16%)	6–10 yy	17/33	dmft = 4.51	76*	71*	38% SDF at baseline (G-B)	5% NSSF at baseline. (G-A)	Participants and examiner	1 mo.	93.42% (71/76)	94.37% (67/71)
											3 mo.	84.21% (64/76)	88.73% (63/71)
											6 mo.	78.95% (60/76)	80.28% (57/71)
											12 mo.	71.05% (54/76)	77.46% (55/71)

Vollú et al. (2019) [41]	34 (8.82%)	33 (21.21%)	2–5 yy	41/26	dmft = 6.72	65	53	30% SDF at baseline	ART	No	3 mo.	100.0% (65/65)	96.15% (50/52)
											6 mo.	89.06% (57/64)	91.67% (44/48)
					ICDAS						12 mo.	88.71% (55/62)	95.56% (43/45)
					5 = 87.7%								
					6 = 12.3%								
Fung et al. (2018) [37]	444 † (10.36%)	444† (9.68%)	3–4 yy †	519/369 †	dmft = 3.85 †	837	847	38% SDF at baseline and repeated every 12 mo. (G3) and every 6 mo. (G4)	12% SDF at baseline repeated every 12 mo. (G1) and every 6 mo. (G2)	Participants and examiner		G3	G1
											24 mo.	46.36% (172/371)	34.25% (124/362)
											30 mo.	49.06% (182/371)	36.74% (133/362)
												G4	G2
											24 mo.	61.68% (227/368)	41.38% (168/406)
											30 mo.	60.65% (222/366)	39.60% (160/404)

n = number; yy = years; M = male; F = female; dmft = decayed, missing and filling primary teeth index; ICDAS = International Caries Detection and Assessment System; SDF = silver diamine fluoride; ART = atraumatic restorative treatment; NaF = sodium fluoride; NSSF = 5% Nano silver incorporated sodium fluoride; mo. = months. * Data after drop out; † data for whole sample including anterior and posterior teeth.

**Table 3 ijerph-19-12917-t003:** Main characteristics of the studies not included in meta-analysis (results rounded up to 1 decimal when 2 decimals were not available).

Author (Year)	N-Participants (% Dropout)			Age YY (Range or Mean)	M/F	Caries at Baseline (Mean)	Intervention			Blinding	Outcomes	Findings			
Daga et al. (2020) [36]	G1	G2	G3	5–8	No Data	No Data	G1	G2	G3	No	Mean active caries (n)		G1	G2	G3
	16 (0.00%)	16 (6.25%)	16 (12.50%)				38% SDF at baseline and at 1-2-3 mo.	38% SDF at baseline and every 3 mo.	38% SDF at baseline and every 6 mo.			0 mo.	2.56	2.25	2.12
												6 mo.	0.43	0.62	1.06
												12 mo.	0.31	0.53	1.35
												Significantly reduction on mean active caries in all groups (*p* = 0.01) and statistically significant differences at 12 mo. among G1 to G3			
Gao et al. (2020) [42]	G1		G2	3–4	No Data	dmft (5.91)	G1		G2	Participants and examiner	Caries arresting rate (%)		G1		G2
	535 (16.45%)		535 (19.06%)				25 % AgNO3 followed by 5 % NaF varnish at baseline and every 6 mo.		38% SDF followed by placebo varnish at baseline and every 6 mo.			6 mo.	41.3%		38.7%
												12 mo.	62.4%		60.0%
												18 mo.	64.1%		62.4%
												24 mo.	68.6%		66.5%
												30 mo.	70.6%		68.9%
												Odds ratio (Ref.: lower posterior teeth)			
												Upper anterior teeth			6.55
												Upper posterior teeth			1.50
												Lower anterior teeth			23.37
												Tooth location significantly related with caries arresting (*p* < 0.01): carious lesions in anterior teeth more likely to be arrested.			
Zhi et al. (2012) [43]	G1	G2	G3	3.8 ± 0.6	82/79	dmft (5.1)	G1	G2	G3	No	Caries arresting rate (%)		G1	G2	G3
	71 (15.49%)	69 (14.49%)	72 (13.9%)				38% SDF at baseline and every 12 mo.	38% SDF at baseline and every 6 mo.	GI			6 mo.	31.5%	43.3%	31.3%
												12 mo.	37.0%	53.0%	28.6%
												18 mo.	77.2%	82.9%	73.1%
												24 mo.	79.2%	90.7%	81.8%
												Odds Ratio (Ref.: posterior teeth)			
												Anterior 5.55			
												Carious lesions in anterior teeth more likely to be arrested (*p* < 0.01)			
Llodra et al. (2005) † [38]	G1		G2	6.29 ± 0.48	229/223	dmfs (3.55)	G1		G2	No	Caries arresting rate (%)		G1		G2
												Primary teeth			
	225 (20.00%)		227 (14.98%)				38% SDF at baseline and every 6 mo.		No treatment			36 mo.	97%		48%
												Significant differences between the groups in mean new decayed surfaces (*p* < 0.01). Significantly more surfaces with inactive caries in G1 (*p* < 0.05).			

N = number; yy = years; M = male; F = female; dmft = decayed, missing and filling primary teeth index; dmfs = decayed, missing and filling surfaces in primary teeth index; SDF = silver diamine fluoride; AgNo3 = silver nitrate; GI = low viscosity, high fluoride-releasing glass ionomer material; mo. = months. † Data included primary teeth (only canines and molars) and permanent teeth (only first molars).

## Data Availability

The data of the current research study are available upon request to the corresponding author.

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
