# Peer review of "Silver Diamine Fluoride (SDF) Efficacy in Arresting Cavitated Caries Lesions in Primary Molars: A Systematic Review and Metanalysis"

_ijerph, 2022, doi:10.3390/ijerph191912917_

Round 1

Reviewer 1 Report

It is a relevant, interesting and well-conducted study. I only have one observation for introduction section. It is to contextualize the justification for conducting the study,  it should be noted that although it was developed in Japan since the late 1960s and used for several decades before its approval in 2014 by Food and Drug Administration, and after this, its use and research has boomed.

Author Response

Dear Reviewer 1,

Thank you for helping us in improving the manuscript.

Please find below our answers to your comments/suggestions.

  • It is a relevant, interesting and well-conducted study. I only have one observation for introduction section. It is to contextualize the justification for conducting the study,  it should be noted that although it was developed in Japan since the late 1960s and used for several decades before its approval in 2014 by Food and Drug Administration, and after this, its use and research has boomed.

Line 60-63 has been changed in “SDF has been used for decades since the late 190s in Japan, China, Brazil and Argentina at concentrations ranging from 10% to 38%. In 2014, after its approval by the Food and Drug Administration as a desensitising agent, its use and scientific interest exploded in the US and recently in Europe.”

Reviewer 2 Report

SDF concentration should rther focus on 38% SDF rather than 30% used in the manuscript since 8 papers used 38% SDF and only 1 paper used 30% (ln 235)

Study selection (ln 207) mentioned '18 papes were discarded' whereas Fig 1 showed 19 studies excluded, 19 was the correct number 

Table 1, 2, 3 all the references number cited in the column Authors (year) was serious error and unorganized such as Abdellatif et al 2021 cited 32 whereas the references list shown 35. These errors had an serious impact on the references cited in the manuscript thereafter (ln 216 219 220 260 etc)

Risk of Bias Assessment from applicator of SDF was classified that 'it was not specify by whom (36)' (ln 260) which was then classified as 'one as high risk (36)'  (ln 269). This might be mis-classification since the un-mentioned applicator or operator could be dental personnel and the risk would then be low. 

Discussion (ln 308-9) '2 studies were excluded from meta-analysis (ref 42 43)', this statement was not clear since these 2 studies were included in all Tables already, otherwise they must be excluded from Study selection Fig 1 rather

(ln 326-330) the first and second sentences were synchronized but quite contrasted with the last sentence with no clear explanation

Conclusion 38% SDF should be used (ln 345)

References suggested for overall check carefully to be related to the cited references in Table Fig and text   

Author Response

Dear Reviewer 2,

Thank you for helping us in improving the manuscript.

Please find below our answers to your comments/suggestions.

  • SDF concentration should rther focus on 38% SDF rather than 30% used in the manuscript since 8 papers used 38% SDF and only 1 paper used 30% (ln 235)

SDF concentration has been changed to 38% from 30% in the abstract and discussion sections

  • Study selection (ln 207) mentioned '18 papes were discarded' whereas Fig 1 showed 19 studies excluded, 19 was the correct number 

Figure 1 has been corrected with the correct number

  • Table 1, 2, 3 all the references number cited in the column Authors (year) was serious error and unorganized such as Abdellatif et al 2021 cited 32 whereas the references list shown 35. These errors had an serious impact on the references cited in the manuscript thereafter (ln 216 219 220 260 etc)

Reference in the tables were corrected. Also, all references matching in the text were verified

  • Risk of Bias Assessment from applicator of SDF was classified that 'it was not specify by whom (36)' (ln 260) which was then classified as 'one as high risk (36)'  (ln 269). This might be mis-classification since the un-mentioned applicator or operator could be dental personnel and the risk would then be low. 

Even without taking this factor into account, as correctly pointed out, using rob2 as a risk of bias assessment, given the overall characteristics of the study, the 'high risk' was determined by several bias, therefore does not change for the mentioned study.

  • Discussion (ln 308-9) '2 studies were excluded from meta-analysis (ref 42 43)', this statement was not clear since these 2 studies were included in all Tables already, otherwise they must be excluded from Study selection Fig 1 rather

The two studies in question were included in the qualitative analysis but not in the meta-analysis because the raw data necessary in order to include the studies in the meta-analysis were not divided by teeth type. Odds ratio based on teeth type was reported instead in these two studies.

  • (ln 326-330) the first and second sentences were synchronized but quite contrasted with the last sentence with no clear explanation

From the results of the meta-analysis, SDF when applied several times is more effective than the ART technique, however, as shown in Figure 2, the results are never statistically significant (please refer to lines 324-325). Lines 368-372 have been modified to better clarify.

  • Conclusion 38% SDF should be used (ln 345)

See comment to the first observation

  • References suggested for overall check carefully to be related to the cited references in Table Fig and text

the references were carefully checked

Reviewer 3 Report

Suggestions:

Inclusion of "diammine" in search to factor for different terminologies- would expect further studies. 

List of excluded studies and reasons for excluding? perhaps in a table/appendix

If any studies used potassium iodide

Inclusion criteria of teeth treated in the included studies would also be helpful (primary molars, anteriors characteristics etc)

Author Response

Dear Reviewer 3,

Thank you for helping us in improving the manuscript.

Please find below our answers to your comments/suggestions.

  • Inclusion of "diammine" in search to factor for different terminologies- would expect further studies. 

The keyword 'diammine' was added to the search strings used for PubMed and Scopus. It was already present in the Embase strings. PubMed results were updated with 4 new articles excluded after title abstract evaluation (lines 207-208). Figure 1 was updated. Supplementary file Table S2 was updated.

  • List of excluded studies and reasons for excluding? perhaps in a table/appendix

List of excluded studies and reasons for excluding are already in the table/appendix. Please see Table S2, Studies excluded after title and abstract screening and Table S3, Studies excluded after full-text reading

  • If any studies used potassium iodide

No study used potassium iodide. Line 304 “No studies have used potassium iodide after SDF application” was added

  • Inclusion criteria of teeth treated in the included studies would also be helpful (primary molars, anteriors characteristics etc)

Inclusion criteria are listed together with teeth characteristics, please refer to paragraph 2.1. However, lines 106-107 were modified in “Type of Intervention applied: SDF applied in active dentin cavitated lesions in primary molars (first and second molars)” to better clarify

Reviewer 4 Report

1.  ines 37-38:  " ... lesions and these ..."  -->  lesiona, and these

2.  lines 207-209 & Table 1:  Consider rewriting lines 207-209 and removing Al-Nerabieah ey al from Table 1.

3.  Line 230  " ... ART ..."  Define this here.  Remove the definition from Table 2.

4.  Table 2 needs a heading.

Author Response

Dear Reviewer 4,

Thank you for helping us in improving the manuscript.

Please find below our answers to your comments/suggestions.

  • ines 37-38:  " ... lesions and these ..."  -->  lesiona, and these

the sentence was modified according to review suggestion

  • lines 207-209 & Table 1:  Consider rewriting lines 207-209 and removing Al-Nerabieah ey al from Table 1.

Following the suggestion, lines were changed in “Nine studies were finally included in this systematic review [35–44]. From the ten studies initially considered eligible, one was excluded due to bias in data reporting (Table 1)[44]. All the papers, except one [35], were published between 2018 and 2021”. Table 1 was modified accordingly.

  • Line 230  " ... ART ..."  Define this here.  Remove the definition from Table 2.

Art definition was inserted in the text (lines 243-244) but was not removed from the table to make the table self-contained and understandable as provided in the guidelines for authors.

  • Table 2 needs a heading.

Table 2 has a heading. Please refer to lines 252-253

Round 2

Reviewer 2 Report

The revised manuscript was more carefully checked especially the references The authors had tried hard to the comments from the reviewers 

However there was some minor on ln 90 (30% SDF) and ln 346 ( ( (=38%) should be 38% SDF 
